# Upregulated Apelin Signaling in Pancreatic Cancer Activates Oncogenic Signaling Pathways to Promote Tumor Development

**DOI:** 10.3390/ijms231810600

**Published:** 2022-09-13

**Authors:** Carline Chaves-Almagro, Johanna Auriau, Alizée Dortignac, Pascal Clerc, Hubert Lulka, Simon Deleruyelle, Fabrice Projetti, Jessica Nakhlé, Audrey Frances, Judit Berta, Véronique Gigoux, Daniel Fourmy, Marlène Dufresne, Anne Gomez-Brouchet, Julie Guillermet-Guibert, Pierre Cordelier, Bernard Knibiehler, Ralf Jockers, Philippe Valet, Yves Audigier, Bernard Masri

**Affiliations:** 1Institut des Maladies Métaboliques et Cardiovasculaires, INSERM U1048, Université de Toulouse, UPS, Toulouse III, 31432 Toulouse, France; 2INSERM ERL1226, CNRS UMR 5215, Université de Toulouse, UPS, Toulouse III, 31432 Toulouse, France; 3Centre de Recherches en Cancérologie de Toulouse, INSERM, CNRS, Université Paul Sabatier, Université de Toulouse, 31037 Toulouse, France; 4Department of Pathology, IUCT-Oncopole, 31100 Toulouse, France; 5Department of Tumor Biology, National Koranyi Institute of Pulmonology, 1121 Budapest, Hungary; 6Institut Cochin, INSERM U1016, CNRS UMR 8104, Université Paris Cité, 75014 Paris, France; 7RESTORE, UMR 1301-Inserm 5070-CNRS EFS, Université de Toulouse, 31100 Toulouse, France

**Keywords:** pancreatic ductal adenocarcinoma, apelin, APJ, G protein-coupled receptor, signaling, oncogenes

## Abstract

Despite decades of effort in understanding pancreatic ductal adenocarcinoma (PDAC), there is still a lack of innovative targeted therapies for this devastating disease. Herein, we report the expression of apelin and its receptor, APJ, in human pancreatic adenocarcinoma and its protumoral function. Apelin and APJ protein expression in tumor tissues from patients with PDAC and their spatiotemporal pattern of expression in engineered mouse models of PDAC were investigated by immunohistochemistry. Apelin signaling function in tumor cells was characterized in pancreatic tumor cell lines by Western blot as well as proliferation, migration assays and in murine orthotopic xenograft experiments. In premalignant lesions, apelin was expressed in epithelial lesions whereas APJ was found in isolated cells tightly attached to premalignant lesions. However, in the invasive stage, apelin and APJ were co-expressed by tumor cells. In human tumor cells, apelin induced a long-lasting activation of PI3K/Akt, upregulated β-catenin and the oncogenes c-myc and cyclin D1 and promoted proliferation, migration and glucose uptake. Apelin receptor blockades reduced cancer cell proliferation along with a reduction in pancreatic tumor burden. These findings identify the apelin signaling pathway as a new actor for PDAC development and a novel therapeutic target for this incurable disease.

## 1. Introduction

Pancreatic ductal adenocarcinoma (PDAC) is one of the most lethal cancers, with an overall five-year survival rate of less than 8% for diagnosed patients. The survival of patients has not improved substantially over nearly 40 years; thus, PDAC represents the eighth worldwide cause of mortality by cancer [1]. It is now well established that highly invasive PDAC arises from local precursor lesions called pancreatic intraepithelial neoplasia (PanIN) [2]. Accumulation of genetic mutations in these premalignant lesions (KRAS, p53, INK4A/ARF and SMAD4) has been implicated in pancreatic cancer progression. However, the earliest detectable mutation, which is the driving force of this pathology, is the mutation of the KRAS gene found in premalignant lesions. To date, only 15–20% of patients are considered to be candidates for surgical treatment. As of today, FOLFIRINOX (leucovorin calcium (folinic acid), fluorouracil, irinotecan hydrochloride and oxaliplatin) and gemcitabine plus nab-paclitaxel are the two common first-line therapies for metastatic adenocarcinoma of the pancreas, which have all improved survival outcomes [3,4]. Despite this slight improvement, identification of new targets and development of novel treatment modalities are an urgent need for this disease.

Apelin is a bioactive peptide identified in 1998 as the endogenous ligand of the orphan G protein-coupled receptor (GPCR) APJ, also called angiotensin receptor-like 1 [5,6]. The apelin gene encodes for a 77 amino acids preproprotein, which generates after the maturation of different fragments, including apelin 36, apelin 17, apelin 13 and pyroglutaminated apelin 13 [7]. At the cellular level, apelin receptor is mainly coupled to Gi proteins [8,9]; thus, apelin fragments inhibit forskolin-induced cAMP production, but also promote phosphorylation of ERKs, Akt, AMPK and p70S6 kinase as well as β-arrestins recruitment and APJ internalization [8,9,10,11].

Both APJ and its cognate ligand apelin are widely present in the central nervous system and in the periphery (heart, blood vessels, adipose tissue, muscle and endocrine pancreas). The first characterized role of apelin signaling was its cardiovascular function (revue [10]). Thus, apelin regulates heart contractility, where it acts as a potent inotropic agent [12]. Moreover, apelin is an angiogenic factor, fostering proliferation, migration of endothelial cells [11,13], promoting the extension of the vascular network [14]. In addition to its cardiovascular benefits, the apelin–APJ system plays a central role in energy metabolism, enhancing glucose disposal in skeletal muscle and adipose tissue in steady state conditions in mice and improving glucose tolerance through increased glucose utilization in obese and insulin-resistant mice [15].

As far as glucose metabolism is concerned, apelin expression has been described in both α and β cells of islets of Langerhans in pancreas. Through its direct action on β cells, a high dose of apelin inhibits glucose-induced insulin secretion [16,17]. Selective APJ deletion in pancreatic islet cells impairs glucose tolerance and glucose-stimulated insulin secretion. Moreover, these animals present reduced islets size, density and β-cell mass [18]. Accordingly, it clearly establishes a stimulatory role for the apelin–APJ signaling axis in the regulation of pancreatic islet homeostasis and energy metabolism. In animal models of acute (AP) and chronic pancreatitis (CP) induced by caerulein injection, apelin and APJ expression is upregulated in the pancreas. In this pathological context, apelin treatment reduces AP- and CP-induced elevations of NF-κB activation in the pancreas and so decreases the inflammatory as well as fibrosis responses by reducing neutrophil recruitment and pancreatic stellate cells activity [19,20].

Using a cancer profiling array, a comparative analysis of apelin gene expression in normal and tumor tissues revealed an overall upregulation of apelin genes in one-third of human tumors [21]. These data were supported by various studies demonstrating an overexpression of apelin and its receptor at the protein level in glioblastoma, oral cancer, colon adenocarcinoma and non-small-cell lung cancer [22,23,24,25]. Interestingly, apelin expression levels in serum, as well as in tumors, positively correlated with tumor progression and decreased overall survival [22,23,26]. Angiogenic, lymphangiogenic and lymph node metastasis properties of the apelin secreted by cancer cells to promote tumor growth and metastasis is now well established [21,22,24,27,28]. However, the function of apelin on tumor cells remains elusive. Among all the different cancers studied by cancer profiling arrays, the extent of apelin upregulation observed in pancreatic cancer reached a 3.5-fold mean increase with the highest frequency (5/7) [21], suggesting that apelin signaling could play an important role in this disease.

In the current study, we demonstrate the involvement of apelin signaling in pancreatic tumor growth and our data unravel a new mechanism of action of apelin to promote tumor growth. Thus, we show that apelin and its receptor are expressed from premalignant lesions to the invasive stage of PDAC. On the pancreatic tumor cell level, the internalization of APJ induced by apelin is the basis for activation of a PI3K/Akt/GSK-3 pathway. Apelin induces an upregulation of β-catenin and the oncogenes c-myc and cyclin D1, as well as glycolysis-related gene expression, and promotes cancer cell proliferation and migration. Of interest, the inhibition of the apelin receptor by shRNA expression in pancreatic tumor cells significantly slows down cancer cell proliferation. More importantly, it reduces tumor burden in a murine orthotopic model of PDAC, highlighting a role of this signaling pathway on pancreatic cancer progression. Altogether, our results establish apelin signaling as a novel and promising therapeutic target for this devastating disease.

## 2. Results

### 2.1. Apelin and APJ Are Differentially Expressed in Normal Pancreas

As previously reported [16], immunostaining for apelin was detected in islets of Langerhans and in acinar cells in a normal mouse pancreas (Figure 1a, left panel). However, no labeling was observed in the ductal cells. Apelin staining was completely lost in the pancreatic sections from apelin knock-out mice (Figure 1a, right panel). Regarding the apelin receptor, its pattern of expression was restricted to islets of Langerhans (Figure 1b, left panel), labeling which was lost when the immunogenic peptide was co-incubated with the antibody (Figure 1b, right panel).

The same expression profile for apelin and APJ was found in the human pancreas (Figure A1). Specificity of the apelin receptor antibody was also confirmed by Western blot on lysates from U2OS stably expressing hAPJ, pretreated or not with tunicamycin, an inhibitor of N-linked glycosylation, as previously described [29] (Figure A2a). As well, APJ was detected by immunofluorescence either at the plasma membrane of those cells or internalized in the complex with β-arrestin 2-YFP when cells were stimulated with apelin (Figure A2b).

To further determine the identity of apelin and APJ-positive islets cells, double immunostaining for apelin or its receptor with the main islet’s hormones (insulin, glucagon and somatostatin) was performed. As previously described [16], in all specimens studied, apelin was co-expressed with insulin in beta cells and with glucagon in alpha cells (Figure 1c, top panels). Regarding the apelin receptor, we identified the same pattern of expression (Figure 1c, bottom panels).

### 2.2. Apelin and APJ Are Expressed in Human and Murine Premalignant and Malignant Pancreatic Lesions

Given the high frequency of apelin gene upregulation in human pancreatic cancer [21], we next determined the expression pattern of the apelin peptide and its receptor in human pancreas sections from 49 patients diagnosed with PDAC. Whereas apelin and APJ were not expressed in normal pancreatic ductal cells in normal mouse and human pancreas (Figure 1a and Figure A1), this couple ligand–receptor was over-expressed in PanINs and PDAC (Figure 2 and Figure 3). More precisely, apelin was expressed in premalignant and malignant lesions including PanINs and PDAC, as well as in islets of Langerhans (Figure 2a,c). In PanIN lesions, apelin staining was both diffuse in the basolateral part of the cells as well as organized as granular patches, mostly localized in a supranuclear position of epithelial cells in well-differentiated glandular structures (Figure 2d). However, expression level decreased, and labeling was diffuse in the cytoplasm of tumor cells in poorly differentiated tumors (Figure 2g). For its part, the expression pattern of APJ was similar to that observed for apelin: expression in PanINs, PDAC and islets of Langerhans (Figure 2b,e). Nevertheless, staining revealed both an intracellular and a membrane localization of APJ in well-differentiated adenocarcinoma (Figure 2f). In poorly differentiated adenocarcinoma, APJ was localized into vesicles which were distributed throughout the cytoplasm of the tumor cells (Figure 2h).

Among 49 samples of human PDAC tissues, apelin and APJ were found expressed in 98% and 82% of cases, respectively, suggesting involvement of this signaling pathway in tumor growth (Table 1). Moderate labeling was observed for apelin and APJ at the different stages of pancreatic cancer as well as in PanINs. Evaluation of association between apelin/APJ and clinicopathological features of PDAC patients did not reveal any significant correlation between apelin or its receptor expression and gender, age or tumor stage (Table 1). To further estimate the impact of apelin and APJ expression levels on the overall survival, we took advantage of a pancreatic adenocarcinoma survival database (TCGA; http://genomics.jefferson.edu/proggene/), (accessed on 9 July 2019). Kaplan–Meier analysis did not show any correlation of apelin or its receptor expression level with overall patient survival (Figure A3).

To define more precisely the spatiotemporal expression of apelin and APJ during pancreatic carcinogenesis, we analyzed their protein expression in an engineered mouse model of PDAC. The LSL-KrasG12D/+; Pdx1Cre/+ mouse model, referred to hereafter as KC, developed the entire compendium of precursor ductal lesions with a slight proportion developing invasive, metastatic adenocarcinoma after a long latency (>1 year) [30], whereas LSL-KrasG12D/+; LSL-p53R172H/+; Pdx-1-Cre (named KPC) was the more aggressive mouse model. In KC mice, apelin was detected in PanIN lesions from low grade (PanIN 1 and PanIN 2) to high grade (PanIN 3), as well as in islets of Langerhans (Figure 3a). Apelin receptor was also found expressed in the endocrine portion of the pancreas. However, APJ was not co-expressed with apelin in PanIN lesions but instead it was localized in isolated cells tightly attached to PanINs or scattered within the ductal epithelium in higher grade lesions (Figure 3b). The same APJ labeling was visualized in some PanIN 1 from human PDAC (Figure A4a). Nevertheless, at the invasive stage, APJ was localized in tumor cells both at the membrane and in the cytoplasm, as observed in human PDAC sections (Figure 3c). The same pattern of expression for both apelin and APJ was observed in the KPC mouse model (Figure A4b).

### 2.3. Apelin Induces Different Signaling Pathways in Human Pancreatic Tumor Cells

To investigate the tumor cell function of the apelin signaling pathway, we first analyzed the endogenous expression of APJ and apelin in different human pancreatic cancer cell lines (MiaPaCa-2, Capan 1, Panc 1 and BxPC 3). Using real-time PCR, APJ was found to be expressed in all tumor cell lines tested, albeit at low levels, but similar to those obtained in human umbilical vein endothelial cells (HUVEC), described to endogenously express apelin receptor [11] (Figure A5a, left panel). Even though we were not able to confirm those data at the protein level using different approaches such as Western blot, enzyme-linked immunosorbent assay or immunofluorescence, stimulation of pancreatic cancer cell lines with exogenous apelin induced the phosphorylation of extracellular signal-regulated kinases (ERKs), as well as Akt, revealing APJ expression (Figure A5b).

We then determined the apelin gene expression in pancreatic cancer cell lines in comparison with HUVECs, which are described to express apelin [24]. We obtained the highest and significant expression level in MiaPaCa-2 cells which presented a six-fold higher expression when compared to that of other cell lines (Figure A5a, right panel). Whereas the apelin secreted by HUVEC cells was quantifiable, we were not able to confirm apelin secretion in the culture medium of the pancreatic cancer cell lines, maybe due to the limit of sensitivity of the kit. Thus, MiaPaCa-2 cells represented the more suitable cell model to define more precisely the apelin signaling function in pancreatic cancer cells since apelin and APJ are co-expressed at the higher level in those cells as observed in human PDAC and mouse models of PDAC.

In MiaPaCa-2 cells, apelin treatment induced a transient phosphorylation of ERKs, p70S6 kinase and its target substrate, the ribosomal protein S6, with a maximal effect between 5 and 15 min (Figure 4a). More interestingly, MiaPaCa-2 cell treatment with apelin caused a sustained increase in Akt and Glycogen Synthase kinase 3 (GSK-3) α and β phosphorylation, which lasted for at least 3 h (Figure 4b). Interestingly, apelin stimulation also generated a time-dependent upregulation of β-catenin protein level (Figure 4b).

To validate that the effects of apelin observed on MiaPaCa-2 cells were dependent on APJ activation, we next abolished the APJ expression using four different nonoverlapping human APJ shRNA (shAPJ 1, shAPJ 2, shAPJ 3 and shAPJ 4) and a nontargeted shRNA (ShScr) as negative control (Figure A6a). Efficiency of shAPJ 3 and shAPJ 4 localized in the coding sequence was validated in U2OS stably expressing APJ and β-arrestin 2-YFP by Western blot (Figure A6b), with Akt phosphorylation stimulated by apelin (Figure A6c), as well as co-internalization of APJ with β-arrestin 2-YFP induced by apelin, as previously described [31] (Figure A6d).

In MiaPaCa-2 cells expressing scramble shRNA (ShScr), apelin induced Akt phosphorylation at the same level as what was obtained in wild-type MiaPaCa-2 cells. On the other hand, the stable expression of the four different human APJ shRNA entirely inhibited apelin-induced Akt activation (Figure 4c). Interestingly, it is worth noting that basal Akt phosphorylation levels in cells stably expressing the different APJ shRNA were weaker than in wild-type MiaPaCa-2 cells or cells stably expressing scrambled shRNA. This difference could be explained by the endogenous apelin secretion that would act in an autocrine/paracrine loop to promote Akt phosphorylation, effect lost in cells deleted for APJ.

### 2.4. Transient ERKs Phosphorylation Induced by Apelin Depends on G-Protein Pathway Whereas Long Lasting Akt Stimulation Relies on APJ Internalization

Our results establish a dual signaling pathway activated by apelin with a transient ERKs and p70S6K phosphorylation and, on the other hand, a long-lasting Akt activation and GSK-3 inhibition. Accordingly, we first determined whether these two transduction cascades depended on Gi/o proteins activation. Pretreatment of MiaPaCa-2 cells with pertussis toxin fully abrogated the ERKs and Akt phosphorylation induced by apelin treatment for 15 min (Figure 5a). Interestingly, Gi/o protein inhibition did not modify apelin-induced Akt activation when cells were stimulated for 3 h (Figure 5b). We next analyzed whether Akt phosphorylation induced by short term apelin treatment was dependent on ERKs signaling pathway. Pretreatment of MiaPaCa-2 cells with a MEK inhibitor, PD098059, did not modify the Akt phosphorylation status induced by apelin (Figure 5c). Accordingly, these results showed that short-term stimulation of MiaPaCa-2 cells with apelin induced two distinct transduction cascades, ERKs/MAPK and Akt, each of them being dependent on Gi/o protein activation. On the other hand, sustained Akt phosphorylation induced by apelin did not rely on a Gi/o protein-dependent pathway.

Activation of Akt is very often linked to the upstream stimulation of Phosphoinositide 3-kinases (PI3K), and PI3K/Akt signaling is one of the most commonly deregulated signaling pathways in cancer. Hence, we next analyzed the involvement of PI3K in persistent Akt activation stimulated by apelin. As shown in Figure 5d, phosphorylation of Akt was repressed by the pan-PI3K inhibitors Wortmannin and LY294002. To further dissect the PI3K isoforms involved in apelin functions, we used isoform-selective inhibitors of the four class IA PI3K, as previously described [32]. Among the different compounds used, only A66S and AS-252424, specifically targeting PI3Kα and γ, respectively, abolished the apelin-induced Akt phosphorylation, while all PI3K inhibitors had an inhibitory action on basal phosphorylation of Akt (Figure 5e).

Finally, we next investigated whether long-lasting Akt phosphorylation induced by apelin could rely on APJ internalization. MiaPaCa-2 cell pretreatment with concanavalin A (100 µg/mL) fully inhibited apelin-induced Akt phosphorylation (Figure 5f). Likewise, the expression of a dominant-negative dynamin-1 cDNA (DynK44A) abrogated the Akt phosphorylation induced by apelin, clearly demonstrating that PI3K/Akt pathway activation occurred through APJ endocytosis (Figure 5g).

### 2.5. Secreted Apelin by Tumor Cells Induced Pancreatic Cancer Burden

Apelin-induced β-catenin stabilization suggested that apelin could display mitogenic properties. Thus, we first analyzed the expression level of oncogenes known to be modulated by β-catenin. Stimulation of MiaPaCa-2 cells with apelin induced a time-dependent increase of c-myc and cyclin D1 protein expression levels (Figure 6a). With those oncogenes being tightly associated with tumor progression, we next analyzed the cellular function of apelin on these cells. As expected, apelin stimulation promoted cancer cell proliferation when compared to untreated control cells (Figure 6b). Similarly, apelin was able to promote MiaPaca-2 cell migration (Figure 6c).

With apelin and APJ being co-expressed by tumor cells both in human pancreatic cancer and PDAC mouse models, we next wondered whether endogenous apelin secreted by pancreatic tumor cells could promote the same effects. For this purpose, we compared the proliferation rate of wild-type MiaPaCa-2 cells (WT) with those stably expressing the ShScr or the four different APJ shRNAs. Interestingly, whereas MiaPaCa-2 WT and ShScr cells proliferated at the same speed, a complete loss of APJ expression significantly reduced their proliferation rate (Figure 6d). These data suggest that the endogenous apelin secreted by tumor cells would promote cell proliferation in an autocrine/paracrine way.

To gain insight into the underlying mechanism by which apelin mediated those cell functions, we assessed apelin activity on glucose metabolism. Apelin treatment significantly increased glucose uptake in the MiaPaCa-2 cells such as insulin which was used as control (Figure 6e). Interestingly, the apelin effect was accompanied by the upregulation of hexokinase II protein, which lasted at least 3 h (Figure 6f). These data implicated a role of apelin in energy metabolism in PDAC. Consistently, hexokinase II as well as glucose transporter Glut 1 expression were significantly upregulated by apelin stimulation, whereas expression level of the enzymes involved in the hexosamine pathway, pentose phosphate pathway and glutaminolysis remained unchanged (Figure A7).

Based on these data, we finally investigated the effect of apelin signaling inhibition on pancreatic tumor growth. We developed an orthotopic pancreatic tumor mouse model by inoculating into the pancreas of SCID mice, wild-type MiaPaCa-2 cells or cells expressing ShScr or shAPJ 3. Whereas the size of the tumors from MiaPaCa-2 cells expressing scramble shRNA and wild-type MiaPaCa-2 cells were similar, loss of APJ expression MiaPaCa-2 cells restrained tumor development (Figure 6g). These results clearly point out that apelin signaling would participate in tumor growth and that its inhibition can significantly reduce pancreatic tumor burden.

## 3. Discussion

Pancreatic ductal adenocarcinoma is an aggressive and devastating disease belonging to the incurable family of solid cancers. Despite decades of effort and medical advances in understanding this deadly malignancy, there is still a lack of innovative targeted therapies. In this study, we identified the apelin signaling pathway as a new regulator of pancreatic tumor growth.

In a normal pancreas, we detected apelin in alpha and beta cells of islets of Langerhans, as well as acinar cells, as already published [16]. A strong immunostaining was also observed in preneoplastic lesions in PDAC mouse models and well-differentiated human adenocarcinoma. This overexpression of apelin peptides extends our previous results which were obtained at the mRNA level in human pancreatic cancer [21]. In regards to the apelin receptor, we detected its expression only in alpha and beta cells of islets of Langerhans in a normal pancreas, which corroborates previous data showing its mRNA expression in isolated islets of Langerhans [16,17] and the ability of apelin to inhibit glucose-induced insulin secretion both in isolated islets of Langerhans and INS-1 clonal beta cells [16,17]. Moreover, APJ expression in alpha cells was confirmed by the capacity of exogenous apelin to activate ERK1/2 and Akt on the hamster glucagonoma INR1-G9 cell line [33] (Figure A8).

Unlike apelin, we did not detect APJ in PanIN tumor cells in premalignant lesions in the KC and KPC mouse models. However, co-expression of apelin and APJ was observed in malignant lesions in both human and mouse models, suggesting that apelin signaling could act on pancreatic tumor cells through an autocrine/paracrine loop, as demonstrated in colon adenocarcinoma [25]. Interestingly, even though APJ was not expressed in PanIN tumor cells, we detected its expression in isolated cells within, or in the vicinity of, PanIN lesions.

The same pattern of expression of apelin and APJ in alpha and beta cells was detected in islets of Langerhans in human adenocarcinoma as well as KC and KPC mouse models. Such a protein expression profile suggests a role for apelin as a paracrine and/or autocrine messenger within the islets.

Moreover, compelling evidence supports the role of islets and their hormones in PDAC [34]. Thus, pancreatic islets regulate glucose metabolism and the growth of surrounding cancer cells—mediated predominantly through β-cell-derived insulin [35]. Likewise, this secreted insulin could positively regulate apelin expression in cancer cells (as described for adipocytes [36]) and act concomitantly with insulin to promote the proliferation and migration of pancreatic cancer cells.

Our data clearly establish that the co-expression of apelin and APJ by pancreatic tumor cells fosters tumor growth, whereas loss of the apelin receptor dramatically reduces the tumor burden in murine orthotopic xenograft experiments. However, we did not observe any correlation between apelin or APJ expression and clinicopathological features of PDAC patients. Similarly, apelin and APJ expression did not seem to correlate with the overall survival prognosis of PDAC patients. This discrepancy can be explained by the small amount of biopsied tissue and the number of patients analyzed. On the other hand, large-scale genomic and transcriptomic data and genetically engineered mouse models show that the oncogenic Kras-PI3K pathway is the major oncogenic driving force in this cancer [37,38]. Kras mutation is the most frequent driving mutation of pancreatic cancer (>90%) and one of the main direct effectors of Kras is the PI3K signal. Because GPCR-dependent signaling contributes to the activation of these enzymes in synergy with oncogenic Kras [39], apelin/APJ could contribute to sustaining this signal.

Increased activation of PI3K signals is one of the most frequent occurrences in all human cancers [40] and fifty percent of pancreatic cancers present an increased activation in PI3K signaling, which correlates with a poor prognosis [41]. Among the different class IA PI3K isoforms, p110α is required for PDAC induced by oncogenic K-Ras [37]. Moreover, p110γ was also found to be overexpressed both in human PDAC and in pancreatic cancer cell lines, playing a pivotal role in pancreatic cancer cell proliferation and tumor burden [42,43]. Our data demonstrate that throughout APJ internalization, the Akt activation pathway induced by apelin relies on both p110α and γ in a redundant manner. To promote long-lasting Akt activation induced by apelin, one well-described mechanism for GPCRs relies on the formation of an intracellular multifunctional protein complex. Internalized GPCRs/β-arrestins complexes can form signalosomes, which are very dynamic in time and subcellular localization, allowing direct recruitment of PI3Ks to β-arrestins, thus permitting modulation of their activity. Such a mechanism has been described for PAR-2 (protease-activated receptor 2), which can promote PI3K activity through a Gαq/Ca2+-dependent pathway and it inhibits PI3K activity through a direct association of β-arrestins 1 and 2 with p110α [44], as well as direct inhibition of p110β by β-arrestin 2 [45]. Formation of such multimeric signaling complexes in cancer cells as well as endothelial cells have been reported to mediate tumor growth, migration, invasion and metastasis in several cancers, including colorectal, breast and ovarian [46,47,48]. In this context—β-arrestins 1 and 2 internalizing with APJ following apelin stimulation [31]—those scaffold proteins could participate with apelin-induced p110α and γ activation to promote PI3K/Akt signaling and pancreatic tumor burden.

Pancreatic tumor cell stimulation by apelin induces inhibitory phosphorylation of GSK-3, a direct downstream Akt effector, and positively regulates β-catenin, c-myc and cyclin D1 protein levels. Those upregulations can be mediated at different levels, such as transcriptional and translational, but can also stabilize the proteins through inhibition of their degradation. In the context of a stressful environment involving nutrient deprivation and hypoxia, as in PDAC, while conventional cap-dependent translation is generally excessively reduced, an alternative translation mechanism depending on internal ribosome entry sites (IRESs) is set up. Interestingly, β-catenin, c-myc and cyclin D1 mRNAs possess IRES structures and the modulation of their expression in PDAC could rely on an IRES-dependent mechanism, as previously demonstrated [49,50]. On the other hand, β-catenin is a specific target of GSK-3, and its phosphorylation by GSK-3 leads to its degradation by the ubiquitin–proteasome pathway. On the contrary, stabilization of β-catenin leads to its nuclear translocation in order to promote the transcription of target genes such as oncogenes c-myc and cyclin D1 [51]. Accordingly, further experiments will be necessary to more precisely define the mechanism involved in the upregulation of those oncogenes through apelin signaling in pancreatic tumor cells.

Cancer cells rewire their metabolism and energy production networks to facilitate survival, proliferation and invasion. Tumor cells undergo metabolic reprogramming through several regulatory pathways such as HIF-1α, PI3K/Akt or c-myc, resulting in an overexpression of the glucose transporters (GLUT1, GLUT2 and GLUT4) found expressed in PDAC [52,53], as well as in the upregulation of several glycolytic genes [54]. Consistent with this important role of glucose metabolism in pancreatic cancer growth, we identified GLUT1 and HK2 as two of the metabolic genes upregulated by apelin that were previously implicated in PDAC [55,56].

Our study provides the first evidence of the involvement of the apelin signaling pathway in pancreatic cancer development through the upregulation of oncogenes and glycolysis-related gene expression, thus promoting tumor cell growth and migration. On the contrary, inhibition of apelin signaling drastically reduced pancreatic tumor development. Collectively, these findings provide new mechanistic insights into pancreatic tumor burden and establish apelin signaling as a novel and promising therapeutic target. Thus, development of specific inhibitors of this signaling pathway could strengthen the standard medical armamentarium already used for this devastating disease.

## 4. Materials and Methods

### 4.1. Reagents

Antibodies to anti-phospho-p44/42 MAPK (#4370 clone D13.14.4E), anti-phospho-S473 Akt (#4060, clone D9E), anti-phospho-S240/S244 S6 protein (#5364, clone D68F8), anti-phospho-S9/S21 GSK3 alpha/beta (#9331), anti-phospho-T389 p70S6 kinase (#9234, clone 108D2), anti-pS240/244 S6 (#5364, clone D68F8), anti-Akt (#4691, clone C67E7), anti-GSK3 alpha/beta (#5676, clone D75D3), anti-Hexokinase II (#2867, clone C64G5), anti-beta Catenin (#8480, clone D10A8) and HRP-conjugated secondary antibodies were purchased from Cell Signaling Technology (Ozyme, Montigny-le-Bretonneux, France). Antibodies to anti-ERK2 (sc-154, clone C-14), anti-Cyclin D1 (sc-450, clone 72-13G) and anti-c-myc (sc-40, clone 9E10) were from Santa Cruz Biotechnology (Santa Cruz, CA, USA). Antibodies directed against tubulin (#T5168, clone B-5-1-2), glucagon (#G2654, clone K79BB10) and pertussis toxin were from Sigma-Aldrich. Antibody to anti-Insulin (#AR029-5R) was from Biogenex (CA, USA) and the antibody to anti-Somatostatin (#A0566) was from Agilent Dako (Les Ulis, France). Antibodies to anti-apelin and anti-APJ were already described [25]. Horseradish peroxidase (HRP)-conjugated secondary antibodies were from Cell Signaling Technology and Alexa fluor 546 goat anti-rabbit and Alexa fluor 488 goat anti-mouse immunoglobulins were from Invitrogen (Saint Aubin, France). Cy2 anti-guinea pig was from Jackson ImmunoResearch Laboratories (Cambridgeshire, UK). Apelin was from Bachem (Bubendorf, Switzerland).

Wortmannin PD098059 and LY294002 were from Calbiochem Novabiochem Corp. (San Diego, CA, USA). PI3K inhibitors (A66, TGX-221, AS-252424, IC-87114) were from Selleckchem (Houston, TX, USA).

Scramble ShRNA and shRNA directed against human apelin receptor were from Sigma-Aldrich (St. Louis, MO, USA).

### 4.2. Mice

The LSL-KrasG12D, LSL-p53R172H knock-in (from D. Tuveson, Mouse Models of Human Cancers Consortium Repository, NCI-Frederick) and Pdx1-cre (from D.A. Melton, Cambridge, MA, USA) mice strains were interbred on a mixed background (CD1/SV129/C57Bl6) to obtain compound mutant LSL-KrasG12D; Pdx1-Cre (named KC) and LSL-KrasG12D; LSL-p53R172H; Pdx-1-Cre (named KPC). Seven-week-old female Swiss nude mice were obtained from Charles River. Mice were kept with 12:12 h light–dark cycle and were fed ad libitum.

### 4.3. Cell Culture

MiaPaCa-2 (CRL-1420), BxPC3 (CRL-1687), Capan-1 (HTB-79), Panc-1 (CRL-1469) and U2OS (HTB-96) cell lines were purchased from the American Type Culture Collection (Rockville, MD, USA) and were cultured in Dulbecco’s Modified Eagle Medium (DMEM) supplemented with 10% fetal calf serum (FCS), 2 mM L-glutamine, 100 U/mL penicillin and 100 μg/mL streptomycin (Gibco, Saint-Aubin, France) in a 5% CO_2_ humidified incubator at 37 °C. HUVEC cells were obtained from PromoCell GmbH (Heidelberg, Germany). U2OS cell line stably expressing human apelin receptor (hAPJ) and β-arrestin2-GFP were already described [31]. No mycoplasma-positive cells were used in this work.

To transiently express the dominant negative K44A mutant of Dynamin, MiaPaCa-2 cells were transfected using Lipofectamine 3000 (Thermo Fisher Scientific, Waltham, MA, USA). MiaPaCa-2 cells expressing control hairpins or hairpins targeting APJ were transduced with lentiviral, encoding for secreted lucia luciferase (Invivogen), as described previously [57].

### 4.4. Proliferation and Migration Assay

MiaPaCa-2 cells were plated in 12-well cell culture plates with at a density of 20,000 cells per well. Twenty-four hours later, apelin was added to the wells at a final concentration of 1 µM and then added every 24 h. Cells were detached and counted every 24 h.

Migration was conducted using 8 µm pore size membrane transwell inserts for 12-well cell culture plates (BD Falcon, Le Pont de Claix, France). After 4 h of serum starvation, medium in the upper chamber was replaced, whereas DMEM containing 5% fetal calf serum, with or without 1 µM apelin, was added to the lower chamber for 18 h. Cells that migrated through the filter membrane were dyed with 0.2% crystal violet.

### 4.5. Immunohistochemistry

Immunohistochemistry was performed as previously described [25].

### 4.6. Western Blot Analysis

MiaPaCa-2 and U2OS cell lines were seeded onto 100 mm culture dishes at 1.5 × 106 cells by dish in complete medium. Twenty-four hours later, cells were serum-starved for twelve hours and stimulated at 37 °C with 500 nM apelin at different time points. Cells were lysed on ice, fractionated on SDS-polyacrylamide gels and proteins of interest detected on nitrocellulose membranes, as previously described, using Amersham hyperfilm ECL (Sigma-Aldrich) or ChemiDoc MPTM (BioRad, Hercules, CA, USA) [11]. Membranes were sliced horizontally so that several proteins could be probed on one blot.

### 4.7. Tumor Model

Human PDAC-derived MiaPaCa-2 cells expressing secreted Gaussia luciferase (WT group), or expressing secreted Gaussia luciferase and control shRNA (shScr), or expressing secreted Gaussia luciferase and shRNA directed against hAPJ (shAPJ 3) were injected into the pancreas of anesthetized (with isoflurane 2%) eight-week-old Swiss nude mice (*n* = 8 mice per group), as previously described [57]. After 47 days, luciferase activity was measured in blood samples using QuantiLuc following the instructions provided (Invivogen, Toulouse, France). Tumor growth was then imaged in anesthetized mice (with isoflurane 2%) using IVIS Spectrum (PerkinElmer, Waltham, MA, USA). The mice were then euthanized by cervical dislocation.

### 4.8. Fluorescence Microscopy

U2OS cells stably co-expressing APJ, β-arrestin 2-GFP and either scramble shRNA or shRNA directed against APJ were seeded on polyD-lysine-coated glass slides in 12-well dishes. Twelve hours later, medium was replaced by fresh medium containing no FCS and cells were either stimulated or not with apelin 13-TAMRA (100 nM) for one hour and then fixed for 15 min in 4% paraformaldehyde in PBS. Slides were mounted in fluorescence mounting medium (Dako, Carpinteria, CA, USA) and images were taken using a Zeiss LSM-780 confocal microscope equipped with appropriate laser lines and filter sets for 488 nm and 564 nm for fluorescence imaging. Images were acquired using a 63× objective and digital zoom set to 1.4×. A minimum of 6 images were collected for each sample and then analyzed with Zen browser software (Zeiss, Thornwood, NY, USA).

### 4.9. Glucose Uptake

MiaPaCa-2 cells were plated in 12-well plates at a density of 20,000 cells per well. twenty-four hours later, cells were serum-starved for 24 h and then stimulated with either apelin or insulin at 1 µM for another 24 h. Ten minutes before the end of the run, 2-deoxy-D-[3H] glucose (2-DG) was added at a final concentration of 0.1 mM. Assay was stopped by replacing the medium with 10 mM of PBS-Glucose. The cells were then rinsed with PBS and lysed using 0.05 N NaOH and intracellular 2-DG was counted.

### 4.10. Quantitative Real-Time PCR

Total cellular RNA was isolated from the different cell lines using RNeasy Mini Kit (QIAGEN, Germatown, MD, USA) and reverse transcribed, as previously described [25].

Primer sequences of human apelin and APJ, as well as of metabolic enzymes involved in the different metabolic pathways, have been already described [36,58].

### 4.11. Statistical Analysis

The results are expressed as mean values ± the standard error of the mean (SEM). The sample size (n) reported in each figure legend refers to the number of independently performed biological replicates in the data set. All analyzable data points were included in the statistical analyses. No statistical methods were used to predetermine the sample size. For experimental methods that were highly reproducible, such as the experiment on cells, a minimum of 3 biologic replicates were sufficient to detect effects of the compounds with *p* < 0.05. For experimental methods with greater variability between replicates, such as cell proliferation and migration or tumor growth, at least 7 biologic replicates were required to attain statistical significance between groups with *p* < 0.01. The investigators were not blinded to allocation during experiments and outcome assessment. Graphs and statistical analyses were performed using GraphPad Prism 7 software (GraphPad Software, La Jolla, CA, USA). Gaussian distribution of data and equality of variances were tested with F Test and the D’Agostino–Pearson normality test. Statistical tests performed are indicated in the figure legends. Univariate statistical analysis was performed using the Pearson’s chi-squared test or Fisher’s exact test. The experiments were not randomized.

A more detailed version of methods and additional methodology are included in Appendix A.

## Figures and Tables

**Figure 1 ijms-23-10600-f001:**
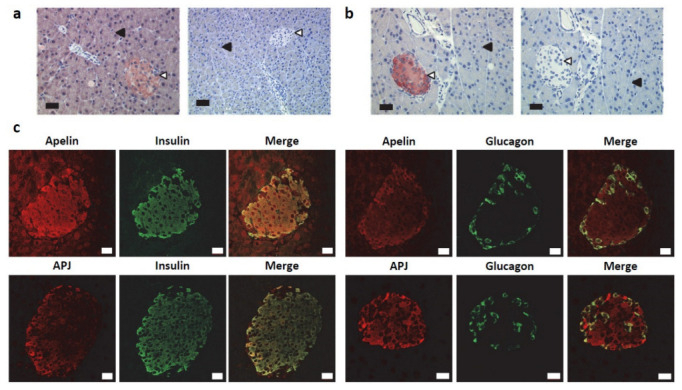
Expression pattern of apelin and apelin receptor in normal murine pancreas. (**a**) Immunohistochemical staining of apelin on a wild-type mouse (left panel, scale bar, 34 µm.) and an apelin knock-out mouse pancreatic section (right panel, scale bar, 67 µm.). Acinar cells (solid triangles) and islets of Langerhans (open triangles) are indicated. (**b**) Immunohistochemical staining of APJ on wild-type mouse pancreatic section. In serial sectioning of pancreas, the staining obtained with the immune serum (left panel) is abolished in the presence of immunogenic peptide (right panel). Scale bar, 34 µm. (**c**) Confocal pictures of double immunostaining for apelin and insulin (top left), apelin and glucagon (top right), APJ and insulin (bottom left) and APJ and glucagon (bottom right). Scale bar, 20 µm. Images are representative of nine independent experiments.

**Figure 2 ijms-23-10600-f002:**
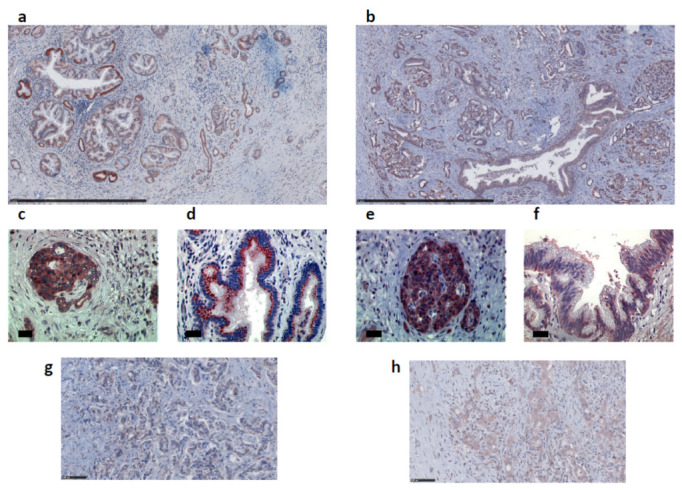
Apelin and APJ are expressed in human premalignant and malignant pancreatic lesions. Representative immunohistochemistry stainings of apelin (**a**) and APJ (**b**) on human PDAC. Scale bars, 1 mm. Apelin (**c**) and APJ (**e**) are expressed in islets of Langerhans. High magnification of apelin (**d**,**g**) and its receptor (**f**,**h**) immunostaining in PanIN lesions (**d**,**f**) and in poorly differentiated pancreatic tumors (**g**,**h**). Scale bars, 34 µm (c–f) or 100 µm (**g**,**h**).

**Figure 3 ijms-23-10600-f003:**
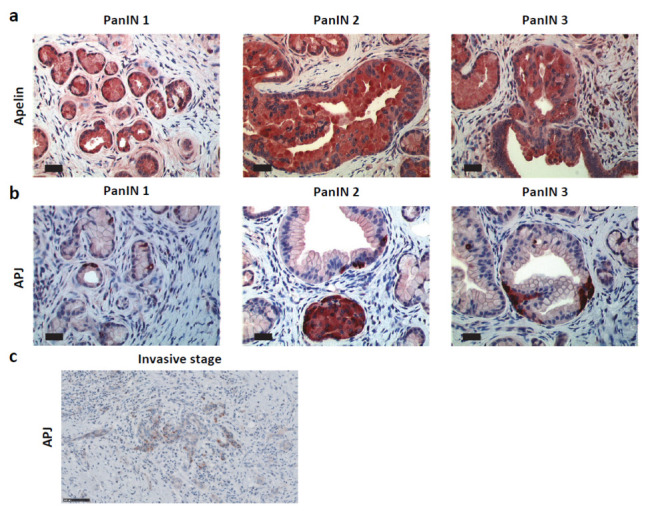
Apelin and APJ are expressed early during pancreatic carcinogenesis. Immunohistochemistry analysis of apelin (**a**) and apelin receptor (**b**) expression from PanIN 1 to PanIN 3 in KC mouse model (*n* = 7). Note the difference in expression pattern between apelin, the expressed by cells comprising PanIN and the APJ expressed by isolated cells surrounding PanIN structures. Scale bar, 34 µm. (**c**) Apelin receptor staining in KC mouse model in invasive stage. Representative image from *n* = 7. Scale bar, 100 µm.

**Figure 4 ijms-23-10600-f004:**
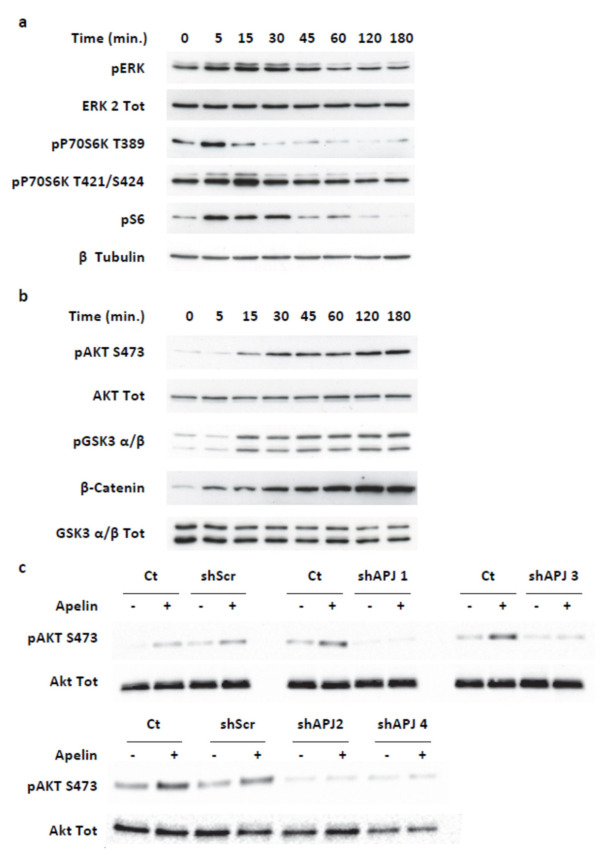
Apelin stimulates transient and long-lasting signaling pathways in the human pancreatic tumor cell line MiaPaCa-2. (**a**) Apelin transiently activates ERK/p70S6 kinase pathways. MiaPaCa-2 cell line endogenously expressing hAPJ were serum-starved for 20 h. After stimulation with 500 nM apelin for the indicated times, they were lysed and analyzed by SDS-PAGE and immunoblotted. Western blots were probed with anti-Thr202/Tyr204 phosphorylated ERK1 and ERK2, anti-ERK2, anti-Thr389 phosphorylated p70S6 kinase, anti-Ser 240/Ser 244 phosphorylated S6 protein and anti-β-tubulin. Representative Western blots from *n* = 4. (**b**) Apelin promotes long lasting phosphorylation of Akt/GSK3 and β-catenin stabilization. Lysates from MiaPaCa-2 cell line, stimulated at different time points with apelin such as in (**a**), were immunoblotted with anti-Ser473, anti-Akt, anti-Ser9/Ser21 phosphorylated GSK3 alpha/beta, anti-GSK3 alpha/beta and anti-β-Catenin. Representative Western blots from *n* = 4. (**c**) Loss of APJ protein expression abolishes apelin response in human pancreatic tumor MiaPaCa-2 cells. MiaPaCa-2 cells stably expressing scrambled shRNA (ShScr) or different shRNA directed against hAPJ (shAPJ 1, shAPJ2, shAPJ 3, shAPJ4) were starved of serum for 20 h and then stimulated with apelin (500 nM) for 3 h. Western blots analysis of phospho-Ser 473 Akt and total Akt were realized. Representative Western blots from *n* = 3.

**Figure 5 ijms-23-10600-f005:**
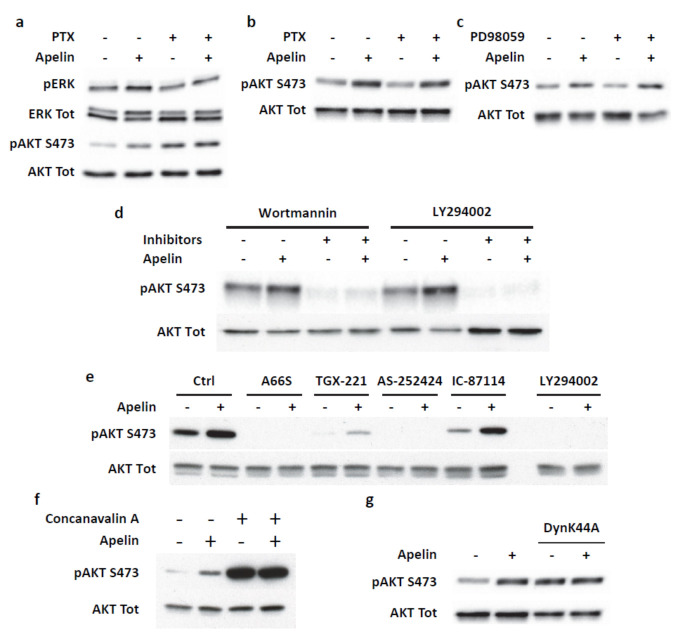
Transient ERK/MAPK stimulation by apelin is dependent on Gi/o proteins, whereas sustained Akt activation is G protein-independent but relies on PI3K α and APJ internalization. (**a**) Analysis of the signaling pathways involved in short-term ERK and Akt activation. MiaPaCa-2 cells were serum-starved and pretreated with pertussis toxin (100 ng/mL) for 20 h and stimulated with apelin 500 nM for 15 min. (**b**–**e**) Analysis of the signaling pathways involved in sustained Akt activation. MiaPaCa-2 cells were serum-starved and pretreated with pertussis toxin (100 ng/mL) for 20 h. (**b**), 1 h with either PD098059 (10 µM) (**c**), Wortmannin (100 nM) and LY204002 (10 µM) (**d**) or A66 (700 nM), TGX-221 (0.5 µM), AS-252424 (100 nM), IC-87114 (10 µM) (**e**), and then stimulated with apelin 500 nM for 3 h. (**f**,**g**) APJ internalization by apelin is a prerequisite to induce Akt activation. MiaPaCa-2 cells were starved of serum for 20 h and pretreated with (**f**) Concanavalin A (100 µg/mL) for 30 min. or transiently transfected with dominant negative K44A mutant of Dynamin (**g**) and stimulated with apelin for 3 h. Representative Western blot from *n* = 3.

**Figure 6 ijms-23-10600-f006:**
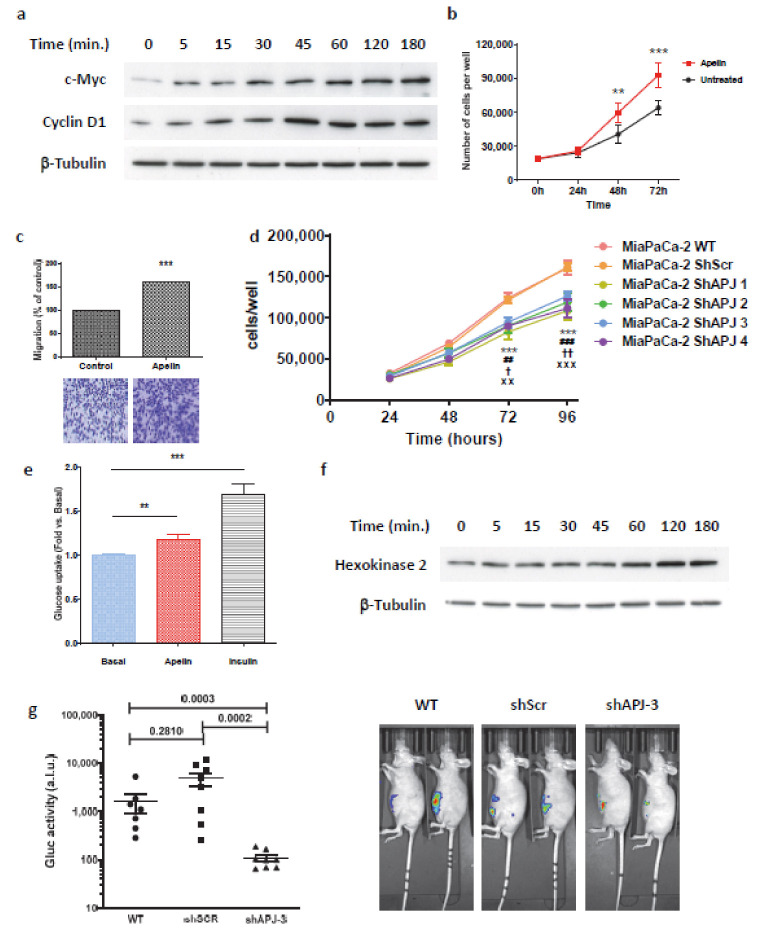
Apelin signaling promotes pancreatic cancer cell proliferation, migration and glucose uptake. Loss of apelin receptor by tumor cells significantly reduces tumor burden. (**a**) Apelin upregulates c-Myc and Cyclin D1 proteins expression. Serum-starved MiaPaCa-2 cells were stimulated with Apelin for the indicated times and Western blot analysis of c-Myc, Cyclin D1 and β-tubulin expressions were realized. Representative Western blot from *n* = 3. (**b**) Apelin treatment induces MiaPaCa-2 cell proliferation (*n* = 4). Data represent means ± SEM. Statistical significance between groups was assessed by using 2-way ANOVA followed by Bonferroni post-test. (**) *p* ≤ 0.01, (***) *p* ≤ 0.001. (**c**) Apelin (1 µM) induces MiaPaCa-2 cell migration. Quantification of migration (*n* = 5). Mean ± SEM; (***) *p* ≤ 0.001, Unpaired Student’s *t*-test. Bottom, representative pictures of crystal violet-colored cells that migrated through the membranes. (**d**) Proliferation of MiaPaCa-2 cells induced by endogenous apelin secretion is fully abolished when MiaPaCa-2 cells stably express an shRNA directed against hAPJ. Data represent means ± SEM. Statistical significance between groups was assessed by using 2-way ANOVA followed by Tukey post-test. *** *p* < 0.001, MiaPaCa-2 ShAPJ 1 vs. MiaPaCa-2 WT or MiaPaCa-2 ShScr; ^##^ *p* < 0.01, ^###^ *p* < 0.001, MiaPaCa-2 ShAPJ 2 vs. MiaPaCa-2 WT or MiaPaCa-2 ShScr; ^†^
*p* < 0.05, ^††^
*p* < 0.01, MiaPaCa-2 ShAPJ 3 vs. MiaPaCa-2 WT or MiaPaCa-2 ShScr; ×× *p* < 0.01, ××× *p* < 0.001, MiaPaCa-2 ShAPJ 4 vs. MiaPaCa-2 WT or MiaPaCa-2 ShScr. (*n* = 4). (e) Apelin (1 µM) promotes glucose uptake in MiaPaCa-2 cells. Insulin (1 µM) was assessed as a positive control. Mean ± SEM; (**) *p* ≤ 0.01, (***) *p* ≤ 0.001, Unpaired Student’s *t*-test. (*n* = 7). (f) Apelin upregulates Hexokinase II protein level in pancreatic tumor cells. Representative Western blot from *n* = 3. (g) Deletion of hAPJ in tumor cells abolishes tumor burden in a murine orthotopic xenograft assay. MiaPaCa-2 cells stably expressing shRNA directed against hAPJ (shAPJ 3) or a control shRNA (shScr) and secreting Gaussia luciferase (Gluc) were injected in the pancreas of Swiss nude mice (*n* = 8 mice per group). Gluc levels were measured in mice serum in the last week of the study period and mice were imaged using IVIS Spectrum. The results are the mean of Gluc activities after 47 days burden (±SEM) and are expressed as Arbitrary Light Unit (A.L.U.). Statistical significance was assessed using Mann–Whitney test.

**Table 1 ijms-23-10600-t001:** Statistical analysis of apelin and APJ proteins expression and clinical characteristics of patients with pancreatic ductal adenocarcinoma (*n* = 49).

		Apelin					APJ				
Characteristics	N° of Cases N = 49	Negative	Weak	Medium	Strong	*p*-Value	Negative	Weak	Medium	Strong	*p*-Value
**Age (years)**						0.24					0.58
< 60	9 (18.4%)	0 (0%)	5 (56%)	4 (44%)	0 (0%)		2 (22.2%)	5 (55.6%)	1 (11.1%)	1 (11.1%)	
≥ 60	40 (81.6%)	1 (2.5%)	17 (42.5%)	10 (25%)	12 (30%)		7 (17.5%)	16 (40%)	16 (40%)	1 (2.5%)	
**Gender**						0.72					0.56
Female	27 (55.1%)	0 (0%)	12 (44.4%)	9 (33.3%)	6 (22.3%)		5 (18.5%)	13 (48.2%)	8 (29.6%)	1 (3.7%)	
Male	22 (44.9%)	1 (4.5%)	10 (45.5%)	5 (22.7%)	6 (27.3%)		4 (18.2%)	9 (40.9%)	8 (36.4%)	1 (4.5%)	
**Adenocarcinoma**	49	1 (2%)	22 (44.9%)	14 (28.6%)	12 (24.5%)		9 (18.3%)	21 (42.9%)	17 (34.7%)	2 (4.1%)	
**Histopathologic grade**						0.84					0.46
WD	27 (55.1%)	1(3.7%)	12 (44.5%)	8 (29.6%)	6 (22.2%)		6 (22.2%)	14 (51.9%)	6 (22.2%)	1 (3.70%)	
MD	14 (28.6%)	0 (0%)	6 (42.9%)	5 (35.7%)	3 (21.4%)		2 (14.3%)	5 (35.7%)	6 (42.9%)	1 (7.1%)	
PD	8 (16.3%)	0 (0%)	5 (62.5%)	1 (12.5%)	2 (25%)		1 (12.5%)	2 (25%)	5 (62.5%)	0 (0%)	
**TNM stage**											
T1	0 (0%)	0 (0%)	0 (0%)	0 (0%)	0 (0%)	0.74	0 (0%)	0 (0%)	0 (0%)	0 (0%)	0.98
T2	8 (16.3%)	0 (0%)	3 (37.5%)	2 (25%)	3 (37.5%)		2 (25%)	4 (50%)	2 (25%)	0 (0%)	
T3	40 (81.6%)	1 (2.5%)	19 (47.5%)	11 (27.5%)	9 (22.5%)		7 (17.5%)	16 (40%)	15 (37.5%)	2 (5%)	
T4	1 (2%)	0 (0%)	0 (0%)	1 (100%)	0 (0%)		0 (0%)	1 (100%)	0 (0%)	0 (0%)	
N0	13 (26.5%)	1 (7.7%)	6 (46.2%)	4 (30.8%)	2 (15.3%)	0.33	3 (23.1%)	7 (53.8%)	2 (15.4%)	1 (7.7%)	0.36
N1	36 (73.5%)	0 (0%)	16 (44.4%)	10 (27.8%)	10 (27.8%)		6 (16.6%)	14 (38.9%)	15 (41.7%)	1 (2.8%)	
**PanINs**						0.64					0.65
Negative	7	7 (100%)					7 (100%)				
PanIN 1	3	0 (0%)	3 (100%)	0 (0%)	0 (0%)		1 (33.3%)	1 (33.3%)	1 (33.3%)	0 (0%)	
PanIN 2	18	1 (5.5%)	5 (27.8%)	9 (50%)	3 (16.7%)		1 (5.6%)	10 (55.6%)	5 (27.8%)	2 (11%)	
PanIN 3	30	0 (0%)	14 (46.6%)	8(26.7%)	8 (26.7%)		3 (10%)	18 (60%)	7 (23.3%)	2 (6.7%)	

Abbreviations: WD, well-differentiated; MD, moderately differentiated; PD, poorly differentiated; PanIN: Pancreatic IntraNeoplasia.

## Data Availability

Data are contained within the article or in the Appendix A.

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
