# Peer review of "Upregulated Apelin Signaling in Pancreatic Cancer Activates Oncogenic Signaling Pathways to Promote Tumor Development"

_ijms, 2022, doi:10.3390/ijms231810600_

Round 1
Reviewer 1 Report
This manuscript describes the upregulation of apelin and its receptor APJ as a potential target in PDAC using shRNA based models to show a reduction in tumor development and growth. I find Figure 3 a little unconvincing to represent co-expression and would prefer either this figure and related discussion removed or additional pictures shown with better co-staining. The statement "no efficient chemotherapy has been developed to improve PDAC survival" is inaccurate as FOLFIRINOX and Gem-Abraxane both improve overall survival and should be corrected.
Author Response
We thank the reviewers for their remarks and thoughtful suggestions. Please find below a point-by-point rebuttal to the reviewer’s comments:
Point 1: This manuscript describes the upregulation of apelin and its receptor APJ as a potential target in PDAC using shRNA based models to show a reduction in tumor development and growth. I find Figure 3 a little unconvincing to represent co-expression and would prefer either this figure and related discussion removed or additional pictures shown with better co-staining.
We understand the point of view of the reviewer regarding the fact that in Figure 3 d to g, only isolated cells are visible on the confocal pictures with the co-labeling and do not clearly show they are part of the PanIN structures. Since those data do not bring any substantial information regarding the main message of the paper and the function of apelin signaling in pancreatic tumorigenesis, we decided to remove Figure 3 d to g as suggested, modified the abstract (line 29), removed the paragraph (lines 208 to 214) describing those results, removed Supplementary Figure S4 C, and modified the discussion (modification line 427 and removal of lines 428 to 432).
Point 2: The statement "no efficient chemotherapy has been developed to improve PDAC survival" is inaccurate as FOLFIRINOX and Gem-Abraxane both improve overall survival and should be corrected.
We modified the paragraph in the introduction in accordance with the comment of the reviewer (lines 49-56).
Reviewer 2 Report
Dr. Chaves-Almagro and the colleagues demonstrated that activation of apelin/APJ axis enhanced malignant potential and promote tumor development. However, no significant difference between expression of apelin/APJ and clinicopathological status (Table 1) as well as survival rate (Fig. A3). More sample size of patients with PDAC should be collected for the analyses, or the criterial should be altered. Indeed, no meaning of distribution of apelin and APJ in islets in this manuscript. The authors should highlight the no expression of apelin/APJ in normal pancreatic ductal cells (Figs. 1a and b; Fig. A1) and overexpression of PanINs and PDAC, rather than the islets’ expression. Meanwhile, in vitro data might be interested and new findings in PDAC cell lines.
Additional major concerns:
Fig.2g and h: The pictures should be altered because these are anaplastic carcinoma with rhabdoid feature, but not poorly differentiated PDAC.
Fig.3c: The duct is typical acinar-to-ductal metaplasia (ADM) with nuclear atypia. The figure should be change to more typical PDAC image.
Fig. A4: These staining patterns might be due to non-specific reaction of secondary antibody. Perform the staining without primary antibody once.
Round 2
Reviewer 2 Report
Almost of issues have been explained and improved in the revised manuscript, but Fig.3c is never accepted for the publication because the revised picture is still showing atypical pancreatic ducts and acinar-to-ductal metaplasia (ADM) with atypia, but not invasive stage of pancreatic ductal adenocarcinoma (PDAC). It might be difficult to distinguish the histology in KC (or KPC) mice for regular pathologists. Indicate a similar picture like Fig. 2g or 2h.
Author Response
We thank the reviewer for his/her remark regarding the figure 3c and the improvement that such modification brings to the conclusions of our manuscript. Please find below a rebuttal to the reviewer’s comment:
Point 1: Almost of issues have been explained and improved in the revised manuscript, but Fig.3c is never accepted for the publication because the revised picture is still showing atypical pancreatic ducts and acinar-to-ductal metaplasia (ADM) with atypia, but not invasive stage of pancreatic ductal adenocarcinoma (PDAC). It might be difficult to distinguish the histology in KC (or KPC) mice for regular pathologists. Indicate a similar picture like Fig. 2g or 2h.
We fully understand the point of the reviewer. As suggested by the reviewer and as done previously for the Figure 2g and h, we replaced Fig. 3c with a more representative one with a larger view in order to observe more easily the expression of APJ protein in invasive stage of pancreatic ductal adenocarcinoma in KC mouse model.